# Experimental Formation and Mechanism Study for Super-High Dielectric Constant AlO_x_/TiO_y_ Nanolaminates

**DOI:** 10.3390/nano13071256

**Published:** 2023-04-02

**Authors:** Jiangwei Liu, Masayuki Okamura, Hisanori Mashiko, Masataka Imura, Meiyong Liao, Ryosuke Kikuchi, Michio Suzuka, Yasuo Koide

**Affiliations:** 1Research Center for Functional Materials, National Institute for Materials Science, 1-1 Namiki, Tsukuba 305-0044, Ibaraki, Japan; 2Applied Materials Technology Center, Technology Division, Panasonic Holdings Corporation, 3-1-1 Yagumo-naka-machi, Moriguchi City 570-8501, Osaka, Japan

**Keywords:** super-high dielectric constant, nanolaminate, ATO, atomic layer deposition, capacitor, double-Schottky contacts

## Abstract

Super-high dielectric constant (*k*) AlO_x_/TiO_y_ nanolaminates (ATO NLs) are deposited by an atomic layer deposition technique for application in next-generation electronics. Individual multilayers with uniform thicknesses are formed for the ATO NLs. With an increase in AlO_x_ content in each ATO sublayer, the shape of the Raman spectrum has a tendency to approach that of a single AlO_x_ layer. The effects of ATO NL deposition conditions on the electrical properties of the metal/ATO NL/metal capacitors were investigated. A lower deposition temperature, thicker ATO NL, and lower TiO_y_ content in each ATO sublayer can lead to a lower leakage current and smaller loss tangent at 1 kHz for the capacitors. A higher deposition temperature, larger number of ATO interfaces, and higher TiO_y_ content in each ATO sublayer are important for obtaining higher *k* values for the ATO NLs. With an increase in resistance in the capacitors, the ATO NLs vary from semiconductors to insulators and their *k* values have a tendency to decrease. For most of the capacitors, the capacitances reduce with increments in absolute measurement voltage. There are semi-circular shapes for the impedance spectra of the capacitors. By fitting them with the equivalent circuit, it is observed that with the increase in absolute voltage, both parallel resistance and capacitance decrease. The variation in the capacitance is explained well by a novel double-Schottky electrode contact model. The formation of super-high *k* values for the semiconducting ATO NLs is possibly attributed to the accumulation of charges.

## 1. Introduction

Oxide insulators with high dielectric constant (*k*) values are important for promoting developments in capacitors, large-capacity memories, and next-generation complementary metal-oxide-semiconductor electronics [1,2,3]. During the past decades, many high-*k* materials, such as Al_2_O_3_, HfO_2_, ZrO_2_, Ta_2_O_5_, TiO_2_, SrTiO_3_, PbZrTiO_3_, etc., have been used instead of low-*k* materials for fabricating high-performance and downscaled electronic devices [4,5,6,7,8,9,10].

Recently, the novel high-*k* oxide materials of Al_2_O_3_/HfO_2_ (AHO) [11,12], Al_2_O_3_/ZnO (AZO) [13], AlO_x_/TiO_y_ (ATO) [14,15,16,17,18], and HfO_x_/TiO_y_ (HTO) [18] nanolaminates (NLs) have been developed for applications in capacitors and field-effect transistors. Compared to a single HfO_2_ insulator, an AHO NL deposited at 120 °C had a lower leakage current density in a capacitor [11]. However, its *k* value (7.4) was not satisfactory. Although annealing at 920 °C could enhance its *k* value up to ~17 [12], this was still lower than that of the HfO_2_ sublayer (~25) [19]. On the other hand, Martinez-Castelo et al. reported that the maximum *k* value for AZO NLs was 9.4 [13]. This was larger than those of both Al_2_O_3_ (~9) [20] and ZnO (~8.5) [21] sublayers. Auciello’s group focused on the depositions of ATO and HTO NLs [14,15,16,17,18]. The maximum *k* value for the ATO NLs was more than 1000, which was much larger than those of the Al_2_O_3_ (~9) [20] and TiO_2_ (~100) [22] sublayers. Meanwhile, that for the HTO NLs (380) was also much higher than the *k* values for the HfO_2_ (~25) [19] and TiO_2_ (~100) [22] sublayers.

For AHO NLs, both Al_2_O_3_ and HfO_2_ sublayers are insulators. However, either the ZnO or TiO_y_ sublayer in the AZO, ATO, and HTO NLs is an intrinsic *n*-type semiconductor [23,24]. Therefore, the insulator/insulator-type AHO NLs reveal *k* values in between the two sublayers. The insulator/semiconductor-type AZO, ATO, and HTO NLs have *k* values higher than both sublayers. One possible explanation for the higher *k* values of the insulator/semiconductor-type NLs is the Maxwell-Wagner effect [14,15,25]. During the current’s passing through the insulating and semiconducting layers, the carrier relaxation times in these two materials are quite different, which leads to the charges accumulated at the interfaces of the two sublayers. These accumulated charges could contribute to the increase in *k* values under an external ac field.

Among all the above NLs, the ATO NLs have exhibited larger capacitance and higher *k* value [14,15,16,17,18]. Some researchers have also made efforts in clarifying their mechanical and optical properties [26,27]. In this study, we carried out the following work to deepen the understanding of ATO NLs: (i) Investigating the effects of experimental conditions, such as ATO NL deposition temperature, number of ATO interfaces, and TiO_y_ content in each ATO sublayer, on the electrical properties of the metal/ATO NL/metal capacitors. (ii) Studying the formation mechanism of the super high-*k* values for the ATO NLs from the viewpoint of a double-Schottky electrode contact model.

## 2. Materials and Methods

The fabrication routines for the silicon/Ti/Pt/ATO NL/Ti/Au (where a “/” symbol shows the deposition sequence) capacitors are shown in Figure 1, including Figure 1a a schematic plan view and Figure 1b a schematic cross-sectional view. After cleaning the silicon with acetone, isopropanol, and ultrapure water sequentially, its surface native oxide was removed in a hydrofluoric acid solution (Figure 1a,b-i). The bottom electrodes of Ti/Pt were deposited on the silicon using evaporation equipment (Model No.: RDEB-1206K; R-DEC Corp., Ltd., Ibaraki, Japan) (Figure 1a,b-ii). The thicknesses of the Ti/Pt were 10/150 nm. The ALO NLs were deposited to cover the entire surface of the silicon/Ti/Pt using an atomic layer deposition (ALD) technique (Model No.: AD-230LP; Samco Inc., Kyoto, Japan) (Figure 1a,b-iii). Precursors for them were trimethylaluminium, tetrakis(dimethylamino)titanium, and water vapor. Deposition rates for the AlO_x_ were 0.74, 0.86, and 0.95 Å/cycle at the deposition temperatures of 150, 200, and 250 °C, respectively. Those for the TiO_y_ were 0.72, 0.66, and 0.56 Å/cycle, respectively.

The following samples were additionally prepared for our study: (i) An ATO NL layer was deposited for scanning transmission electron microscope (STEM) and energy dispersive X-ray (EDX) measurements. The thicknesses of the ATO NL and an AlO_x_/TiO_y_ sublayer were 54.3 and 1.98/2.01 nm, respectively. Hereafter, 1.98A/2.01T is used to represent the thicknesses of the ATO sublayers. (ii) A single AlO_x_ (23.1 nm), a single TiO_y_ (18.8 nm), and three ATO NLs (0.34A/0.73T, 0.51A/0.53T, and 0.68A/0.33T with each NL’s thickness around 50 nm) were deposited at 200 °C on silicon/Ti/Pt for the Raman measurements. (iii) Three samples with ATO NL and sublayer thicknesses around 50 nm and 0.5A/0.5T, respectively, were deposited at 150, 200, and 250 °C. They were used to clarify deposition temperature effects on the electrical properties of the capacitors. (iv) Three samples (0.51A/0.52T) with ATO NL thicknesses of 51.9, 103.8, and 201.5 nm, respectively, and one sample (2.04A/2.08T) with a thickness of 103.0 nm were deposited at 200 °C for clarifying the effect of the number of ATO interfaces on the electrical properties of the capacitors. The 0.51A/0.52T ATO NLs were also used for the impedance measurement. (v) Five samples (0.34A/0.73T, 0.51A/0.52T, 0.60A/0.39T, 0.68A/0.33T, and 0.77A/0.19T), deposited at 200 °C with each ATO NL thickness at around 100 nm, were employed to clarify the AlO_x_ and TiO_y_ content effects on the electrical properties of the capacitors.

The circular Ti/Au (10/100 nm) metals were evaporated to cover the silicon/Ti/Pt/ATO NL for the top electrodes (Figure 1a,b-iv). The diameter of the Ti/Au electrodes was 200 μm. The ATO NLs were etched using a capacitively coupled-plasma reactive ion etching system (Model No.: RIE-200NL; Samco Inc., Kyoto, Japan) to open the windows for the bottom Ti/Pt electrodes (Figure 1a,b-v). The plasma power, CHF_3_ flow rate, and Ar flow rate were 100 W, 10 sccm, and 40 sccm, respectively. The electrical properties of the silicon/Ti/Pt/ATO NL/Ti/Au capacitors were measured using a MX-200/B prober (Model No.: MX-200/B; Vector Semiconductor Corp., Ltd., Tokyo, Japan) and a B1500A parameter analyzer (Model No.: B1500A; Agilent Technologies Inc., Tokyo, Japan) at room temperature.

## 3. Results

### 3.1. Schematic Diagram for the Capacitor and EDX Mapping for the ATO NL

Figure 2a shows a schematic diagram of the silicon/Ti/Pt/ATO NL/Ti/Au capacitor. The measurement probes contacted with the top Ti/Au and bottom Ti/Pt electrodes for characterizing the electrical properties. Figure 2b,c show the STEM image and EDX mapping for the 1.98A/2.01T ATO NL. The green and red colors represent the aluminum and titanium elements, respectively. It was observed that there are individual multilayers for the ATO sublayers with around 2 nm thickness. Thus, the ATO NLs were confirmed to be deposited successfully by controlling thicknesses using the ALD technique.

### 3.2. Raman Spectra for the ATO NLs

Figure 3a shows the Raman spectra for the single AlO_x_ and TiO_y_ layers. The peaks with the highest intensities of each are at positions of 1534.6 and 618.9 cm^−1^, respectively. Figure 3b shows the Raman spectra for the 0.34A/0.73T, 0.51A/0.52T, and 0.68A/0.33T ATO NLs. With the increase in AlO_x_ content in each ATO sublayer, the peak intensity at 618.9 cm^−1^ decreases greatly. The intensity ratios between the two peaks at the positions of 1534.6 and 618.9 cm^−1^ are 0.93, 1.09, and 1.46, respectively. The shape of Raman spectrum for the AlO_x_-rich 0.68A/0.33T ATO NL is close to that of the single-layer AlO_x_. 

### 3.3. Effects of Deposition Temperature on Electrical Properties of the Capacitors

Figure 4a–d show the leakage current-voltage (*I-V*), capacitance-frequency (*C-f*), capacitance-voltage (*C-V*), and loss tangent-frequency (tanδ*-f*) characteristics of the capacitors with the variation in ATO NL deposition temperatures. The deposition temperatures for the ATO NLs were 150 (black circle line), 200 (red circle line), and 250 °C (green circle line). The voltage changed from −5 V to 5 V for the *I-V* characteristics. With the increments in the ATO NL deposition temperature, the leakage current increased. The leakage currents at 1 V were 2.48 × 10^−6^, 2.02 × 10^−4^, and 4.92 × 10^−3^ A for the capacitors with the ATO NLs deposited at 150, 200, and 250 °C, respectively (Figure 4a). Their resistances can be calculated to be 4.02 × 10^5^, 4.96 × 10^3^, and 2.03 × 10^2^ Ω, respectively.

With the measurement frequency changing from 1 kHz to 1 MHz, the capacitances decreased [Figure 4b]. This is attributed to the existence of series resistances for the capacitors [28]. The maximum capacitance (*C_max_*) values are 0.96, 2.49, and 3.01 μF/cm^2^ for the capacitors with the ATO NLs deposited at 150, 200, and 250 °C, respectively. The *k* value of the ATO NL can be calculated using the following Equation (1):(1)$$k=\frac{Cd}{\varepsilon_{0}}$$
where *d* and *ε*_0_ are the thickness of the ATO NL and the dielectric constant of vacuum (8.85 × 10^−12^ F m^−1^), respectively. The maximum *k* value for the ATO NL deposited at 250 °C was 174.9. This is larger than that for those deposited at 200 and 150 °C, of 146.0 and 55.4, respectively.

With an increase in absolute voltage, the capacitances for all of three capacitors were reduced (Figure 4c). The reduction in capacitance for the ATO NL (150 °C) capacitor was more dramatic than those of the ATO NLs (200 and 250 °C). The tan*δ* was 0.29 at 1 kHz for the ATO NL (150 °C) capacitor (Figure 4d). It was much lower than those of the other two capacitors, of 4.64 and 871, respectively. Therefore, in this section, the ATO NL deposition temperature effects on electrical properties of the capacitors have been clarified. The lower deposition temperature of the ATO NLs can lead to a lower leakage current and higher resistance in the capacitors. The higher deposition temperatures are related to the higher *k* values for the ATO NLs, more dramatic variations for the *C-V* curves, and the larger tan*δ* values.

### 3.4. Effects of the Number of ATO Interfaces on Electrical Properties of the Capacitors

Figure 5a–d show the *I-V*, *C-f*, *C-V*, and tanδ*-f* characteristics of the capacitors with the variations in the NL thickness and the number of ATO interfaces. The sublayer thicknesses are 0.51A/0.52T for the black, red, and green circle lines and 2.04A/2.08T for the blue circle line. The total ATO NL thicknesses were 51.9, 103.8, 201.3, and 103.0 nm and the corresponding number of ATO interfaces were 50, 100, 194, and 25, respectively. With increasing thickness and number of interfaces for the 0.51A/0.52T ATO NLs, the leakage current decreased slightly (Figure 5a). Although the number of ATO interfaces for 2.04A/2.08T NL was only 25, the leakage current for the capacitor was around four orders lower than those for the 0.51A/0.52T ATO NL. Based on the leakage currents at 1 V, the resistances for four capacitors were calculated to be 4.96 × 10^3^, 5.54 × 10^3^, 5.88 × 10^4^, and 1.98 × 10^7^ Ω, respectively. The *C_max_* values for the three 0.51A/0.52T ATO NL capacitors were 2.49, 2.35, and 1.95 μF/cm^2^, which were much larger than that (0.24 μF/cm^2^) of the 2.04A/2.08T one. Their *k* values were calculated to be 146.0, 275.6, 443.5, and 27.9, respectively. The capacitances decreased dramatically for the 0.51A/0.52T ATO NL capacitors with the increments in absolute voltage (Figure 5c). That for the 2.04A/2.08T one was very stable. With the increase in the number of interfaces for the 0.51A/0.52T ATO NLs, the tanδ at 1 kHz decreased from 4.64 to 0.23 (Figure 5d). For the whole region of measurement frequency, the tanδ for the 2.04A/2.08T ATO NL capacitor was very stable and lower than 0.38. It is observed that there are some relationships between leakage current and the other properties of the capacitors. When leakage currents are lower, the *C-V* characteristics are more stable and the tanδ values are lower. These relationships are in good agreement with the situations in the above Section 3.3 and Section 3.4.

### 3.5. Effects of AlOx and TiOy Contents on Electrical Properties of the Capacitors

Figure 6a–d show *I-V*, *C-f*, *C-V*, and tanδ*-f* characteristics for the capacitors with variations in AlO_x_ and TiO_y_ content in each ATO sublayer. Deposition temperature, total thickness, and each ATO sublayer thickness for all the ATO NLs were kept at 200 °C, around 100 nm, and around 1 nm, respectively. The ATO sublayers change from 0.34A/0.73T, 0.51A/0.52T, 0.60A/0.39T, 0.68A/0.33T to 0.77A/0.19T. With the increments in AlO_x_ content and the reduction in TiO_y_ content, the leakage currents for the capacitors decreased from 3.98 × 10^−3^ A to 3.42 × 10^−10^ A (Figure 6a). The resistances can be calculated to increase from 2.51 × 10^2^ Ω to 2.93 × 10^9^ Ω. The *C_max_* values for the five capacitors were 3.64, 2.35, 1.12, 0.35, and 0.09 μF/cm^2^, respectively (Figure 6b). Their *k* values can be calculated to be 418.7, 275.6, 126.3, 39.9, and 10.2, respectively. With the increase in absolute voltage, the capacitances for the 0.34A/0.73T, 0.51A/0.52T, 0.60A/0.39T ATO NL capacitors decreased dramatically (Figure 6c). Those for the 0.68A/0.33T and 0.77A/0.19T ones were somewhat steady. Except for the 0.34A/0.73T ATO NL capacitor, the tan*δ* values for the capacitors were lower than 1.1 in the range of 1 kHz ~1 MHz (Figure 6d). Based on the analysis of results in this section, we can conclude that the higher TiO_y_ content could lead to a higher leakage current, lower resistance, higher *k* value, sharper change for the *C-V* curve, and larger tan*δ* value.

### 3.6. Impedance Properties of the Capacitors

Figure 7a shows the impedance spectra of the three 0.51A/0.52T ATO NL capacitors. They are measured at 0 V in the frequency from 1 Hz to 1 MHz. All of the spectra have semi-circular shapes, which indicates that the equivalent circuit for the Si/Ti/Pt/ATO NL/Ti/Au capacitors is composed of resistances and capacitance [29]. With the ATO NL thickness increasing from 51.9 nm to 201.3 nm, the size of the semi-circular shape increases. This is ascribed to the variations in resistance and capacitance. Figure 7b shows the equivalent circuit for the Si/Ti/Pt/ATO NL/Ti/Au capacitors. It is composed of the series resistor (*R_S_*), parallel resistor (*R_P_*), and capacitance. The *R_P_* and capacitance as functions of voltage derived from the impedance spectra are shown in Figure 7c,d, respectively. With the increments in ATO NL thickness, the *R_P_* increased from 1.7 × 10^2^ Ω, through 3.2 × 10^3^ Ω to 1.2 × 10^4^ Ω at −2 V. With the increase in absolute voltage, the capacitances decreased for all three samples, which is in good agreement with the change tendencies shown in Figure 5c. 

## 4. Discussion

We will discuss the mechanism by which the ATO NL produces the huge dielectric constant in this section. Table 1 summarizes the electrical properties of the Si/Ti/Pt/ATO NL/Ti/Au capacitors. Figure 8a–d show the *k* values of the ATO NLs as functions of the ATO NL deposition temperature, number of ATO interfaces, TiO_y_ content in each ATO sublayer, and resistance for the capacitor, respectively. The open and solid circle spots in Figure 8b represent the 0.51A/0.52T and 2.04A/2.08T ATO sublayers, respectively. It is observed that a higher ATO NL deposition temperature, larger number of ATO interfaces, and higher TiO_y_ content in each ATO sublayer can lead to higher *k* values for the ATO NLs. The highest *C_max_* and *k* values were 3.64 μF/cm^2^ and 443.5, respectively. This *k* value is lower than that of previous reports (~1000) [14,15]. This is possibly attributable to the lower deposition temperature and fewer interfaces of the ATO NLs in this study. With the same sublayer thickness (0.51A/0.52T) and total NL thickness (103.8 nm), the *k* value (275.6) for the ATO NL is lower than the value (380) of the HTO NL [18]. One possible reason is the *k* value of the HfO_2_ is higher than that of the Al_2_O_3_. The lowest equivalent oxide thickness (EOT) value is 0.9 nm. This low value is important to push forwards the development of next-generation complementary metal-oxide semiconductor electronics.

With the increments in resistance, the *k* values have a tendency to decrease (Figure 8d). The leakage currents of the capacitors with ATO NLs deposited at 200 and 250 °C in Figure 4, the 0.51A/0.52T ones in Figure 5, and the 0.34A/0.73T, 0.51A/0.52T, and 0.60A/0.39T ones in Figure 6 are larger than 6.72 × 10^−6^ A. Their resistances were in the range of 2.03 × 10^2^~1.49 × 10^5^ Ω. The resistivity values can be calculated to be in the range of 77.5~4.7 × 10^4^ Ω m for the ATO NLs. They are lower than 10^5^ Ω and are considered as semiconductors (blue region in Figure 8d). On the other hand, the leakage currents for the ATO NL (150 °C) capacitor in Figure 4, the 2.04A/2.08T one in Figure 5, and the 0.68A/0.33T and 0.77A/0.19T ones in Figure 6 are lower than 2.48 × 10^−6^ A. Their resistances and *ρ* values are larger than 4.02 × 10^5^ Ω and 2.5 × 10^5^ Ω m, respectively. These ATO NLs are considered as insulators (green region in Figure 8d).

Based on previous reports [14,15], the Maxwell-Wagner effect was employed to explain the formation of super high-*k* values for the ATO NLs. Because of the relaxation time difference between insulating AlO_x_ and semiconducting TiO_y_ layers, the charges are accumulated at the ATO interfaces as the current passes through them. In addition, the dipole charge stack on the AlO_x_ sublayer predicts a large *k* value if the insulating ATO NL with large resistivity is obtained. With the increments in the number of interfaces in the ATO NLs, the *k* value would also increase. This is in good agreement with the summary result shown in Figure 8b. However, when the number of ATO interfaces is kept at 100 (Figure 8c), the *k* values also change with the variation in AlO_x_ and TiO_y_ contents in each ATO sublayer. Therefore, the Maxwell-Wagner model has some limitations in explaining the super high-*k* values for the ATO NLs.

In this study, we attempted to explain it using the double-Schottky electrode contact model. The major reason is in the decrement tendency of the capacitance while increasing the applied voltage for the ATO NLs with large *k* values. The schematic diagram of the model is shown in Figure 9. For the metal/semiconducting ATO NL/metal capacitors, there are the following relationships for depletion layer width (*x_d_*), capacitance, and applied voltage [30],
(2)$$x_{d}=\sqrt{\frac{2\varepsilon_{0}k(Q_{bi}-V)}{qN_{d}}}$$
(3)$$C=\frac{\varepsilon_{0}k}{x_{d}}=\sqrt{\frac{q\varepsilon_{0}kN_{d}}{2(Q_{bi}-V)}}$$
where *q*, *N_d_*, and *Q_bi_* are the elementary charge (1.6 × 10^−19^ C), donor concentration of ATO NL, and built-in potential, respectively. The *x_d_* is in proportion to the (*Q_bi_* − V). When the applied voltage is positive (+V), the *x_d_* at the bottom Ti/Pt side is enlarged (Figure 9a). When the applied voltage is negative (−V), the *x_d_* is increased at the top Ti/Au side (Figure 9c). Since both *x_d_* values are larger than that at V = 0 V (Figure 9b), an increase in them would lead to a decrease in capacitances in Equation (3). Therefore, with an increase in absolute voltages, the capacitance for metal/semiconducting ATO NL/metal capacitor decreases theoretically. This is in good agreement with the experimental results in Figure 4, Figure 5 and Figure 6c and the fitting results in Figure 7d. The accumulation of charges due to the Schottky electrode contacts possibly leads to the increase in capacitance and the *k* values (>126.3) of semiconducting ATO NLs higher than both AlO_x_ (~9) [20] and TiO_y_ (~100) [22] sublayers. On the other hand, since the samples of ATO NL (150 °C) in Figure 4, the 2.04A/2.08T one in Figure 5, and the 0.68A/0.33T, 0.77A/0.19T ones in Figure 6 are insulators (green region in Figure 8d), they are not suitable for the Schottky electrode contact model. Their *k* values are in the range of 10.5~55.4, which is in between two sublayers. In order to further confirm the mechanism for the huge dielectric constants of the ALD NLs, the following works would be need to be performed: (i) Other contact metals with different work functions would need to be deposited on the ATO NLs to clarify their effects on the dielectric constants. (ii) It should be confirmed whether the AZO and HTO NLs agree with the Schottky electrode contact model.

## 5. Conclusions

In this study, the effects of ATO NL deposition conditions on the electrical properties of the silicon/Ti/Pt/ATO NL/Ti/Au capacitors were investigated. A higher deposition temperature, larger number of ATO interfaces, and higher TiO_y_ content in each ATO sublayer were important for enhancing the *k* values of the ATO NLs. The higher leakage currents or lower resistances of the capacitors were related to the higher *k* values of the ATO NLs, more dramatic variations in the *C-V* curves, and larger tan*δ* values. With the increase in resistances for the capacitors, the ATO NLs varied from semiconductors to insulators. The double-Schottky electrode contact model was employed to explain the variations in capacitances with the increments in absolute voltage. The super-high *k* values for the semiconducting ATO NLs were possibly attributed to the accumulation of charges caused by the Schottky electrode contacts. 

## Figures and Tables

**Figure 1 nanomaterials-13-01256-f001:**
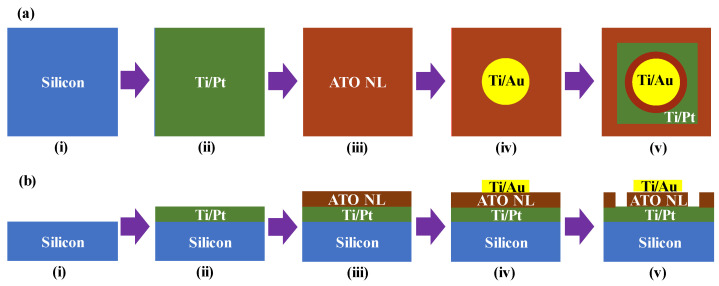
Fabrication routines for the Si/Ti/Pt/ATO NL/Ti/Au capacitors in (**a**) schematic plan view and (**b**) schematic cross-sectional view: (i) silicon substrate cleaning, (ii) Ti/Pt evaporation, (iii) ATO NL deposition, (iv) Ti/Pt evaporation, and (v) ATO NL etching to open the windows for Ti/Pt electrodes.

**Figure 2 nanomaterials-13-01256-f002:**
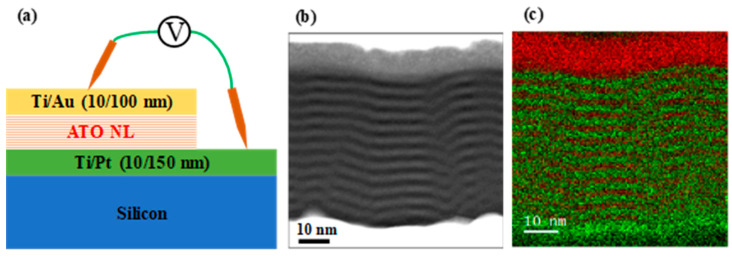
(**a**) Schematic diagram of the Si/Ti/Pt/ATO NL/Ti/Au capacitor. (**b**) STEM image and (**c**) EDX mapping for the 1.98A/2.01T ATO NL layer.

**Figure 3 nanomaterials-13-01256-f003:**
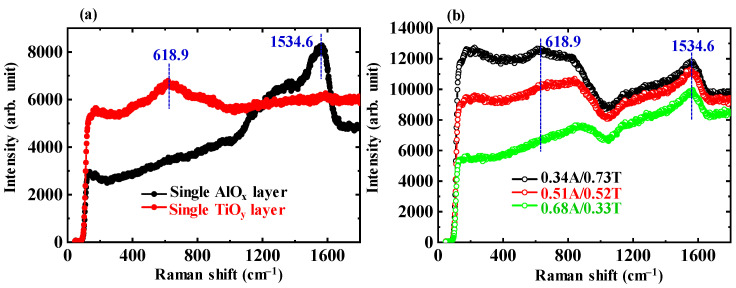
Raman spectra for (**a**) AlO_x_ and TiO_y_ single layers and (**b**) 0.34A/0.73T, 0.51A/0.52T, and 0.68A/0.33T ATO NLs.

**Figure 4 nanomaterials-13-01256-f004:**
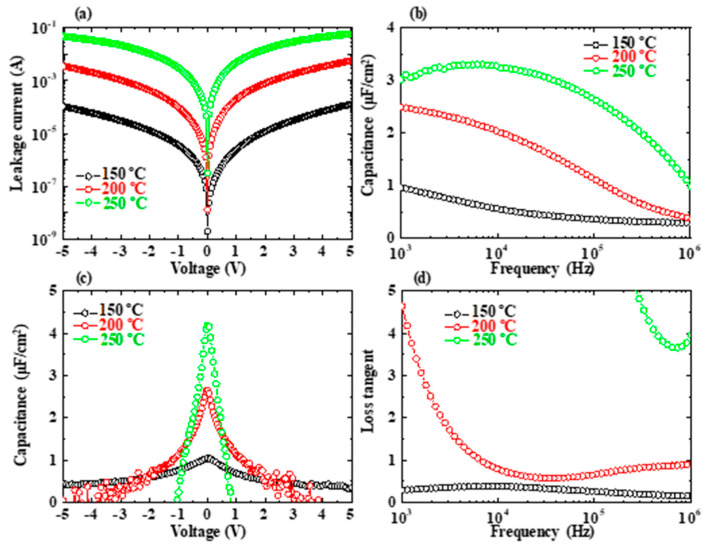
(**a**–**d**) Leakage current-voltage, capacitance-frequency, capacitance-voltage, and loss tangent-frequency characteristics of the Si/Ti/Pt/ATO NL/Ti/Au capacitors with the variation in ATO NL deposition temperature. The ATO NL and sublayer thicknesses are the same at around 50 nm and 0.5A/0.5T, respectively. Their deposition temperatures were 150 (black circle line), 200 (red circle line), and 250 °C (green circle line).

**Figure 5 nanomaterials-13-01256-f005:**
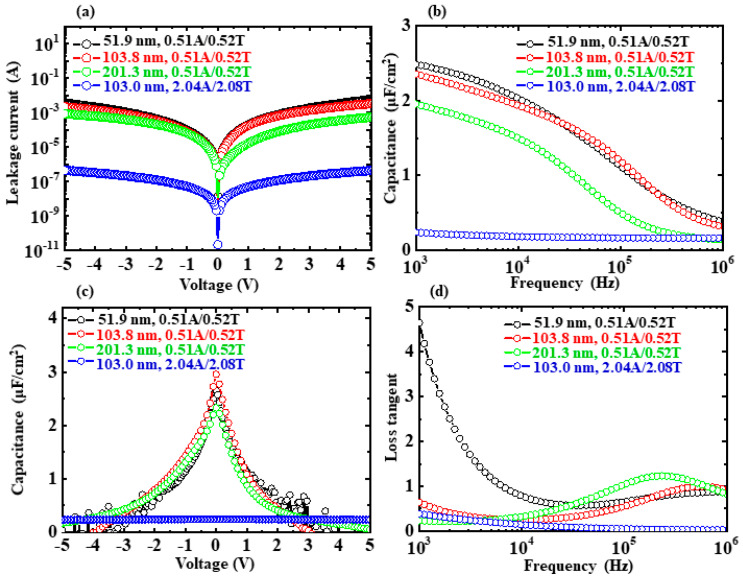
(**a**–**d**) Leakage current-voltage, capacitance-frequency, capacitance-voltage, and loss tangent-frequency characteristics for the Si/Ti/Pt/ATO NL/Ti/Au capacitors with variation in NL thickness and the number of ATO interfaces. The deposition temperature for the four ATO NL layers was the same at 200 °C. The ATO sublayer thicknesses were 0.51A/0.52T (black, red, and green circle lines) and 2.04A/2.08T (blue circle line). The numbers of ATO interfaces for the four samples were 50, 100, 194, and 25, respectively.

**Figure 6 nanomaterials-13-01256-f006:**
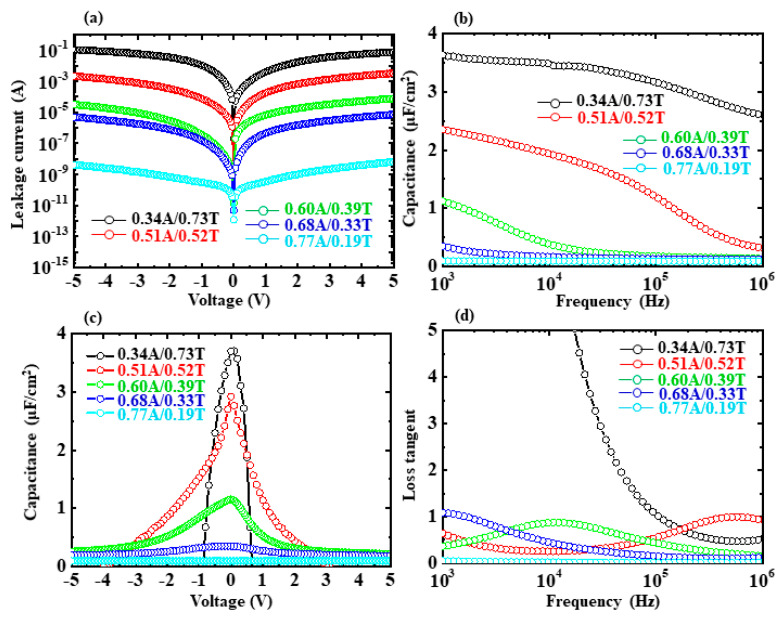
(**a**–**d**) Leakage current-voltage, capacitance-frequency, capacitance-voltage, and loss tangent-frequency characteristics for the Si/Ti/Pt/ATO NL/Ti/Au capacitors with the variation in AlO_x_ and TiO_y_ contents. The deposition temperature, total NL thickness, and each ATO sublayer thickness were kept at 200 °C, around 100 nm, and around 1 nm, respectively. The ATO sublayer thicknesses change from 0.34A/0.73T to 0.77A/0.19T.

**Figure 7 nanomaterials-13-01256-f007:**
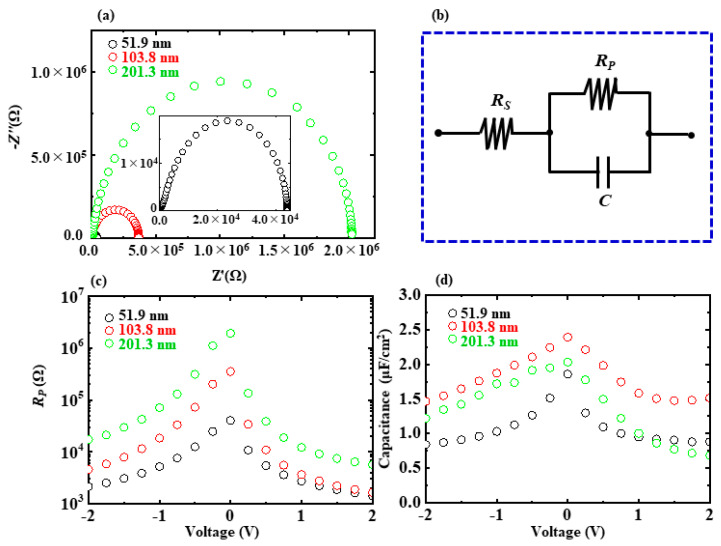
(**a**) Impedance spectra of the Si/Ti/Pt/ATO NL/Ti/Au capacitors measured at 0 V in the frequency from 1 Hz to 1 MHz. The sublayer thicknesses for them are the same at 0.51A/0.52T. The total thicknesses for the ATO NL layers are 51.9 (black circle), 103.8 (red circle), and 201.3 nm (green circle). (**b**) Equivalent circuit for the Si/Ti/Pt/ATO NL/Ti/Au capacitors. (**c**,**d**) *R_p_* and capacitance as functions of voltage derived from the impedance spectra, respectively.

**Figure 8 nanomaterials-13-01256-f008:**
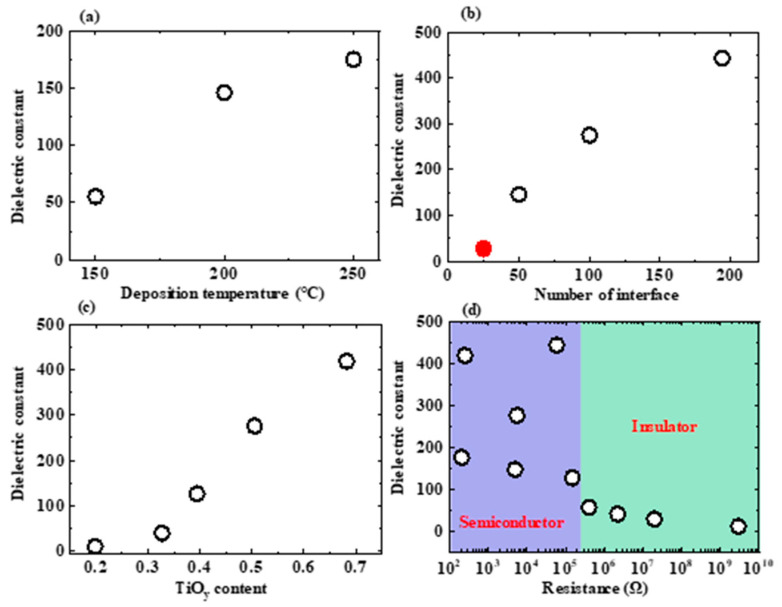
Dielectric constants of the ATO NLs as functions of (**a**) deposition temperature, (**b**) number of ATO interfaces, (**c**) TiO_y_ content, and (**d**) resistance of the capacitor. The open and solid circle spots in Figure 8b represent the 0.51A/0.52T and 2.04A/2.08T ATO NLs, respectively.

**Figure 9 nanomaterials-13-01256-f009:**
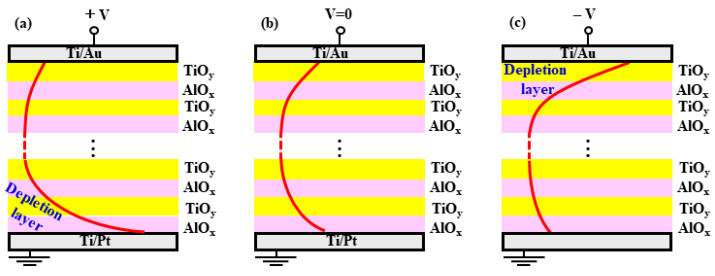
Schematic diagrams of the double-Schottky electrode contact model at a voltage of: (**a**) +V, (**b**) V = 0, and (**c**) −V, respectively.

**Table 1 nanomaterials-13-01256-t001:** Summary of the electrical properties of the Si/Ti/Pt/ATO NL/Ti/Au capacitors.

ATO NL Deposition Temperature (°C)	ATO Sublayer Thickness (nm)	ATO NL Thickness (nm)	Number of ATO Interfaces	Leakage Current (A)@1 V	Resistance (Ω)	*C_max_* (μF/cm^2^)	*k*	EOT (nm)	tanδ @1 kHz
150	0.51/0.50	51.1	50	2.48 × 10^−6^	4.02 × 10^5^	0.96	55.4	3.6	0.29
200	0.51/0.52	51.9	50	2.02 × 10^−4^	4.96 × 10^3^	2.49	146.0	1.4	4.64
250	0.47/0.46	46.9	50	4.92 × 10^−3^	2.03 × 10^2^	3.30	174.9	1.0	871
200	0.51/0.52	103.8	100	1.80 × 10^−4^	5.54 × 10^3^	2.35	275.6	1.5	0.65
200	0.51/0.52	201.3	194	1.70 × 10^−5^	5.88 × 10^4^	1.95	443.5	1.8	0.23
200	2.04/2.04	103.0	25	5.06 × 10^−8^	1.98 × 10^7^	0.24	27.9	14.4	0.38
200	0.34/0.73	101.8	100	3.98 × 10^−3^	2.51 × 10^2^	3.64	418.7	0.9	81.0
200	0.60/0.39	99.8	100	6.72 × 10^−6^	1.49 × 10^5^	1.12	126.3	3.1	0.36
200	0.68/0.33	101.0	100	4.52 × 10^−7^	2.21 × 10^6^	0.35	39.9	9.9	1.08
200	0.77/0.19	100.1	100	3.42 × 10^−10^	2.93 × 10^9^	0.09	10.2	38.3	0.05

## Data Availability

The data presented in this study are available on request from the corresponding author.

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
