# Peer review of "Experimental Formation and Mechanism Study for Super-High Dielectric Constant AlOx/TiOy Nanolaminates"

_nanomaterials, 2023, doi:10.3390/nano13071256_

Round 1

Reviewer 1 Report

Comments to nanomaterials-2309455

The manuscript presents an AlOx/TiOy nanolaminates structure. It shows a very high dielectric constant. There are some comments toward the research.

1.     The fabrication routines of Figure 1(a)(v) does not match with Figure 1(b)(v), please revise it.

2.     In the discussion session, from line 303 to line 305 presents the relation between xd and applied voltage. It is suggested to describe in detail why the applied positive voltage would increase xd of bottom Ti/Pt side? Why the applied of negative voltage would increase xd of the Ti/Au side?

3.     What’s the physical meaning of “ accumulation of depletion charges”? for the depletion layer of a semiconductor means a region with no free charge there, the reader may confuse about the “accumulation of depletion charges”. 

4.     Why the accumulation of depletion charges could increase the value of dielectric constant?

Author Response

Please find the response letter in the attached file. Thank you very much.

Reviewer 2 Report

In this work, the author demonstrates the high C value in the ML scheme MIM structure. The high k/C value is very important for Logic device and DRAM device.  The work is solid with the C-V, Raman, TEM image. I have the minor suggestion that:

1. Could the author comments and plot K-EOT with other's work for the benchmark?

2. Could author comment that this IL can be used on Ge substarte or Si substarte only ?

Author Response

(The authors gave the same response as above.)

Reviewer 3 Report

The printed manuscript describes the study of super-high dielectric constant (k) AlOx/TiOnanolaminates produced by ALD. The electrical properties of the produced nanolaminates were investigated and analysed. It was demonstrated that the super-high values for the semi- 326 conducting ATO NLs were possibly attributed to the accumulation of depletion charges caused by the Schottky electrode contacts. Even though the topic is well studied, the authors provided some significant experimental results Ann explanations that could interest the broad scientific audience. In my opinion, the manuscript may be accepted after some revisions listed below:

- the introduction should be improved: (i) please stress the novelty of the proposed research; (ii) there are a lot of publications about TiO2/AlO3 nanolaminates which the authors have not overviewed and the obtained results are not compared (see for instance: https://doi.org/10.1021/acs.jpcc.5b06745; https://doi.org/10.1016/j.matdes.2016.09.030);

- the obtained results should be compared with other nanolaminates produced by ALD (e.g. HfO2 based nanolaminates); could this methodology be spread to other nanolaminates?

- how did the author control and check the thicknesses of produced layers? Did they use ellipsometry?

- the morphology and phase composition should be studied more profoundly using additional experimental techniques (e.g. XPS, HR-TEM, XRD). What was the crystallinity of produced layers?

- is there any inter diffusion of elements from one layer to another? How it depends on the temperature and the time of operation?

- the future perspective of the proposed/developed nanolaminates should be provided.

Author Response

(The authors gave the same response as above.)

Reviewer 4 Report

The manuscript does not fulfil the aims and scopes of the special  issue  "Nanoscale thin film transitors and application exploration"since NO transistor is reported.The manuscript could  be considered as a regular paper .

Author Response

Dear Reviewer,  

We are truly grateful to your kind suggestion. This manuscript is not related to the field-effect transistors directly. However, the investigation of oxide insulators with high dielectric constant values are important to promote developments of metal-insulator-semiconductor field-effect transistors (MISFETs). This is the reason that the Lead Guest Editor of this Special Issue invited us to submit this kind of paper.

Once again, thank you very much for your kind review.

Sincerely Yours,

Jiangwei Liu, Ph.D

National Institute for Materials Science

Research Center for Functional Materials

1-1 Namiki, Tsukuba, 305-0044, Japan

Tel: +81-29-860-4954

Round 2

Reviewer 3 Report

The authors addressed all my comments 

Reviewer 4 Report

Accepted